# Role of Enzymic Antioxidants in Mediating Oxidative Stress and Contrasting Wound Healing Capabilities in Oral Mucosal/Skin Fibroblasts and Tissues

**DOI:** 10.3390/antiox12071374

**Published:** 2023-06-30

**Authors:** Parkash Lohana, Albert Suryaprawira, Emma L. Woods, Jordanna Dally, Edward Gait-Carr, Nadia Y. A. Alaidaroos, Charles M. Heard, Kwok Y. Lee, Fiona Ruge, Jeremy N. Farrier, Stuart Enoch, Matthew P. Caley, Matthew A. Peake, Lindsay C. Davies, Peter J. Giles, David W. Thomas, Phil Stephens, Ryan Moseley

**Affiliations:** 1Disease Mechanisms Group, Oral and Biomedical Sciences, School of Dentistry, College of Biomedical and Life Sciences, Cardiff University, Cardiff CF14 4XY, UKwoodse1@cardiff.ac.uk (E.L.W.); dallyj2@cardiff.ac.uk (J.D.); nalaidaroos@hstp.gov.sa (N.Y.A.A.); leekydai@gmail.com (K.Y.L.); m.caley@qmul.ac.uk (M.P.C.); matthew.peake@ncl.ac.uk (M.A.P.);; 2Canniesburn Plastic Surgery Unit, Glasgow Royal Infirmary, Glasgow G4 0SF, UK; 3School of Pharmacy and Pharmaceutical Sciences, College of Biomedical and Life Sciences, Cardiff University, Cardiff CF10 3NB, UK; heard@cardiff.ac.uk; 4Cardiff China Medical Research Collaborative, School of Medicine, Cardiff University, Cardiff CF14 4XN, UK; 5Oral and Maxilliofacial Surgery, Gloucestershire Royal General Hospital, Gloucester GL1 3NN, UK; 6Department of Burns and Plastic Surgery, University Hospital of South Manchester, Manchester M23 9LT, UK; 7Cell Biology and Cutaneous Research, Blizard Institute, Barts and the London School of Medicine and Dentistry, Queen Mary University of London, London E1 2AT, UK; 8School of Biology, Natural and Environmental Sciences, Newcastle University, Newcastle Upon Tyne NE1 7RU, UK; 9Department of Microbiology, Tumor and Cell Biology, Karolinska Institutet, Solnavägen 9, Biomedicum, 17165 Solna, Sweden; 10Division of Medical Genetics, School of Medicine, College of Biomedical and Life Sciences, Cardiff University, Cardiff CF14 4XN, UK; gilespj@cardiff.ac.uk; 11Advanced Therapies Group, Oral and Biomedical Sciences, School of Dentistry, College of Biomedical and Life Sciences, Cardiff University, Cardiff CF14 4XY, UK; thomasdw2@cardiff.ac.uk (D.W.T.);

**Keywords:** oral mucosa, skin, fibroblasts, wound healing, oxidative stress, antioxidants, superoxide dismutase 3 (SOD3)

## Abstract

Unlike skin, oral mucosal wounds are characterized by rapid healing and minimal scarring, attributable to the “enhanced” healing properties of oral mucosal fibroblasts (OMFs). As oxidative stress is increasingly implicated in regulating wound healing outcomes, this study compared oxidative stress biomarker and enzymic antioxidant profiles between patient-matched oral mucosal/skin tissues and OMFs/skin fibroblasts (SFs) to determine whether superior oral mucosal antioxidant capabilities and reduced oxidative stress contributed to these preferential healing properties. Oral mucosa and skin exhibited similar patterns of oxidative protein damage and lipid peroxidation, localized within the lamina propria/dermis and oral/skin epithelia, respectively. SOD1, SOD2, SOD3 and catalase were primarily localized within epithelial tissues overall. However, SOD3 was also widespread within the lamina propria localized to OMFs, vasculature and the extracellular matrix. OMFs were further identified as being more resistant to reactive oxygen species (ROS) generation and oxidative DNA/protein damage than SFs. Despite histological evaluation suggesting that oral mucosa possessed higher SOD3 expression, this was not fully substantiated for all OMFs examined due to inter-patient donor variability. Such findings suggest that enzymic antioxidants have limited roles in mediating privileged wound healing responses in OMFs, implying that other non-enzymic antioxidants could be involved in protecting OMFs from oxidative stress overall.

## 1. Introduction

Although oral mucosal and dermal wounds proceed through similar stages of healing, oral mucosal wounds are commonly characterized by limited inflammation, rapid healing and minimal scar formation, in contrast to adult dermal wounds that are usually accompanied by prominent scar formation [1,2]. Similar to regenerative mechanisms in early-gestational fetal skin [3], distinct gene expression and response differences exist between the fibroblast populations residing within the lamina propria of the oral mucosa (OMFs), compared to those within the dermis of skin (SFs). Such contrasting genotypic profiles between OMFs and SFs contribute to the differential wound healing capabilities of oral mucosal and skin tissues, particularly in terms of the superior proliferative, migratory and matrix metalloproteinase (MMP)-mediated extracellular matrix (ECM) remodeling properties of OMFs, closely associated with their “younger” phenotype [4,5,6,7,8,9]. Furthermore, although SF-myofibroblast differentiation induced by pro-fibrotic mediators, such as transforming growth factor-β_1_ (TGF-β_1_), is a pivotal response in facilitating normal wound closure, contraction, pro-fibrotic ECM deposition and scar formation in skin [10], OMFs exhibit lower TGF-β_1_ expression and resistance to TGF-β_1_-driven myofibroblast differentiation compared to SFs, thereby retaining their ‘non-scarring’ phenotype [5,11,12,13].

Expression profiling comparisons between patient-matched OMFs and SFs have enhanced our understanding of the preferential wound healing responses of OMFs at a molecular level through identification of the key genes involved, such as hepatocyte growth factor (HGF) [5,9,13,14,15,16,17]. However, another prominent regulator of normal and pathological wound healing and scarring mechanisms is oxidative stress, being particularly recognized in dermal tissues [18,19,20,21]. Additionally, oxidative stress has further been implicated in the initiation and progression of cancer and other diseases in both oral mucosal and skin tissues [22,23,24,25]. Oxidative stress refers to the balance in reactive oxygen species (ROS) production and cellular antioxidant defense mechanisms. ROS are generated via a wide range of cellular mechanisms, with low ROS levels purported to play important roles in regulating cell signaling and functions [26,27]. However, although tightly regulated enzymic and non-enzymic antioxidant defense mechanisms counteract ROS accumulation, excessive ROS production can cause indiscriminate damage to biomolecules, such as DNA, proteins and lipids, leading to altered cellular functions [18,19,26,28]. Consequently, differences in cellular and tissue susceptibilities to oxidative stress often correlate with their enzymic antioxidant capabilities, most notably superoxide dismutases (SODs) and catalase, including in fibroblasts [29,30,31,32,33]. Thus, imbalances between ROS and antioxidant levels are established to influence dermal wound healing and scarring outcomes [18,19,20,21]. Furthermore, atypical oral mucosal healing with scar formation, such as that clinically manifested during the pre-cancerous, chronic inflammatory condition of oral submucous fibrosis also has oxidative stress as an underlying contributor to disease pathology [34,35].

Despite ever-increasing evidence to implicate oxidative stress as a key regulator of wound healing and scarring, to date, no studies have determined whether differences in oxidative stress responses or enzymic antioxidant capabilities exist between oral mucosal and skin tissues and fibroblasts. Therefore, this study examined the distribution of oxidative stress biomarkers and major enzymic antioxidants between patient-matched oral mucosal and skin tissues, in addition to patient-matched OMFs and SFs, to determine whether reduced oxidative stress or superior enzymic antioxidant capabilities contribute to the preferential healing and reduced scarring properties of the oral mucosa.

## 2. Materials and Methods

### 2.1. Oral Mucosal and Skin Fibroblasts and Tissues

Patient-matched biopsies (6 mm) of normal, non-diseased human buccal mucosa and skin tissues were obtained from adults (*n* = 8) undergoing routine oral surgery procedures at the University Dental Hospital, Cardiff, and Vale University Health Board, Cardiff, UK. Biopsies were collected with informed patient consent and ethical approval by the South East Wales Research Ethics Committee of the National Research Ethics Service (NRES), UK. Oral mucosal and skin biopsies (*n* = 4, patients 1–4) were immediately snap frozen in n-hexane (ThermoFisher Scientific, Lutterworth, UK), floated on liquid nitrogen. Serial cryostat sections (10 μm) were subsequently cut and mounted onto poly-L-lysine (Sigma, Poole, UK) coated microscope slides. The additional oral mucosal and skin biopsies (*n* = 4, patients 5–8) were used for the establishment of patient-matched OMF and SF cultures, as previously described [8,9].

### 2.2. Oxidative Stress Biomarker and Enzymic Antioxidant Immunohistochemistry

Cryosections were fixed in acetone (Sigma) for 15 min, air dried for 10 min and washed in Tris-buffered saline (TBS, ThermoFisher Scientific, 3 × 5 min). Sections were subsequently immunolabeled with a panel of antibodies directed against various oxidative stress biomarkers and enzymic antioxidants. For the detection of carbonyl group formation resulting from protein oxidation, sections were reacted overnight with acid 2,4-dinitrophenylhydrazine (2,4-DNPH) reagent (15 mM 2,4-DNPH dissolved in absolute ethanol containing 1.5% (*v*/*v*) concentrated sulfuric acid) and processed, as previously described [28,36]. All sections were blocked for nonspecific binding with normal serum (Vectorstain Universal Elite ABC Kit, Vector Laboratories, Peterborough, UK) for 20 min. Sections were subsequently incubated with the appropriate primary antibodies for 30 min at room temperature, diluted in 1% bovine serum albumin (BSA, Sigma) in TBS. These included protein carbonyl/oxidized protein contents (rabbit antisera anti-dinitrophenyl antibody, 1:2000, Agilent, Ely, UK), lipid peroxidation/malondialdehyde contents (rabbit IgG polyclonal antibody, 1:500, Autogen Bioclear, Calne, UK), SOD1 (rabbit IgG polyclonal antibody, 1:700, Abcam, Cambridge, UK), SOD2 (rabbit IgG polyclonal antibody, 1:125, Caltag-Medsystems, Buckingham, UK), SOD3 (rabbit IgG polyclonal antibody, 1:1000, Antibody Technology, Scoresby, Australia) and catalase (rabbit IgG polyclonal antibody, 1:500, Abcam). Immunoreactivity was determined using a Vectorstain Universal Elite ABC Kit and a DAB peroxidase kit (Vector Laboratories). Sections were counterstained with hematoxylin (Merck Millipore, Watford, UK) for 30 s and mounted. Tissue sections were visualized by light microscopy (Olympus Provis Digital Microscope, Olympus UK Ltd., Southend-on-Sea, UK), with digital images captured using ACT Digital Photo Software v.2 7.

### 2.3. Oral Mucosal and Patient-Matched Skin Fibroblast Cultures

OMFs and patient-matched SFs were cultured in Fibroblast-Serum-Containing Medium (F-SCM), containing Dulbecco’s Modified Eagle’s Medium (DMEM) and supplemented with L-glutamine (2 mM), non-essential amino acids (×1), antibiotics (100 U/mL penicillin G sodium, 100 μg/mL streptomycin sulfate and 0.25 µg/mL amphotericin B) and 10% fetal calf serum (FCS) (all ThermoFisher Scientific). Cultures were maintained at 37 °C in 5% CO_2_/95% air, with culture medium changed every 2–3 days.

### 2.4. Determination of Endogenous Reactive Oxygen Species (ROS) Generation

Firstly, superoxide radical (O_2_^●−^) generation by patient-matched OMFs/SFs was quantified by cytochrome C reduction [37]. Patient- and passage-matched OMFs and SFs were established in 24-well plates (5 × 10^4^ cells/well) and maintained at 37 °C in 5% CO_2_/95% air in F-SCM for 72 h. At 24 h, 48 h and 72 h, OMFs and SFs were washed (×3) in phosphate buffered saline (PBS) and replenished with serum-free, phenol red-free DMEM (ThermoFisher Scientific), with L-glutamine (2 mM), antibiotics (100 U/mL penicillin G sodium, 100 μg/mL streptomycin sulfate and 0.25 µg/mL amphotericin B) and cytochrome C (80 µM, horse heart type III, Sigma). Cultures were maintained at 37 °C in 5% CO_2_/95% air for 2 h. Culture medium was subsequently removed for the spectrophotometric determination of cytochrome C reduction at 550 nm using a DU 800 UV/Visible Spectrophotometer (Beckman Coulter Ltd., High Wycombe, UK). Remaining OMFs and SFs were treated with 0.05% trypsin/0.53 mM EDTA (ThermoFisher Scientific) and viable cell counts were determined using 0.4% Trypan blue (Sigma). Levels of O_2_^●−^ generation were calculated using a molar extinction coefficient of 21,000 cm/moles/L and corrected for viable cell number.

Further studies focused on the visualization of detectable ROS generation by patient-matched OMFs/SFs, using 2′,7′-dichlorofluorescein diacetate (DCF). Patient- and passage-matched OMFs and SFs were grown to 80–90% confluence in 8-well chamber slides (VWR International, Lutterworth, UK). Cells were subsequently loaded with DCF (10 µM, Sigma) and maintained under darkness at 37 °C in 5% CO_2_/95% air, for 15 min. OMF and SF chamber slides were counterstained with Hoechst nuclear dye stain (Sigma), mounted using Fluor Save Reagent (Merck Millipore) and viewed using a Zeiss Axiovert 200 M Inverted Microscope (Carl Zeiss Ltd., Cambridge, UK). Images were captured and processed using Adobe Photoshop Elements 2018 (Adobe Systems, San Jose, CA, USA). Controls for each patient-/passage-matched OMF and SF culture were also established, consisting of OMF/SF with no DCF, but in the presence of Hoechst stain.

### 2.5. Oxidative DNA Biomarker Detection

Patient- and passage-matched OMFs and SFs were grown to 30–40% confluence in 8-well chamber slides. Oxidative DNA damage, in the form of 8-OHdG levels, was detected using fluorometric OxyDNA Assay Kits (Merck Millipore) per manufacturer’s instructions [38]. OMF and SF chamber slides were counterstained, mounted, viewed images processed, as described above. Controls for each patient-/passage-matched OMF and SF culture were also established, consisting of OMF/SF with PBS instead of the FITC-conjugate, but in the presence of Hoechst stain.

### 2.6. Oxidative Protein Biomarker Detection

Patient- and passage-matched OMFs and SFs were established in T-75 tissue culture flasks (1 × 10^6^ cells/flask) and maintained at 37 °C in 5% CO_2_/95% air in F-SCM, until confluent. Cultures were subsequently washed in PBS (5 mL × 3) and the cell-ECM contents harvested into 2 mL ice-cold, 50 mM Tris-HCl buffer, pH 7.5, containing 5 mM EDTA and 1 mM dithiothreitol (both Sigma). Extracts were sonicated and protein concentrations quantified (Pierce^®^ BCA Protein Assay Kit, ThermoFisher Scientific) according to manufacturer’s instructions. Oxidative protein damage (in the form of protein carbonyl contents) wasdetected using Oxyblot Protein Oxidation Detection Kits (Merck Millipore), with extracts (10 μg protein) derived according to manufacturer’s instructions [36,39].

Samples (20 μL) were subjected to sodium dodecyl sulfate-polyacrylamide gel electrophoresis (SDS-PAGE) under reducing conditions on preformed 10% linear gels (Mini-Protean^®^ Tetra Cell System, BioRad, Hemel Hempstead, UK) and electroblotted onto polyvinylidene difluoride membranes (Hybond™-P; ThermoFisher Scientific), using a Mini Trans-Blot^®^ Electrophoretic Transfer Cell (BioRad) per manufacturer’s instructions. Membranes were blocked with 1% BSA in 0.05% Tween 20 (ThermoFisher Scientific)/PBS, pH 7.2, overnight at 4 °C. Membranes were immuno-probed with primary antibody (rabbit anti-DNPH antibody, in Kit), diluted 1:150 in 0.05% Tween 20/PBS, pH 7.2, for 1 h at room temperature. Protein loading was confirmed by β-actin Loading Control (1:20,000, Abcam). Membranes were washed (×3) in 0.05% Tween 20/PBS, pH 7.2 at room temperature, incubated in secondary antibody (anti-rabbit, horseradish peroxidase-conjugated IgG antibody, raised in goat, in Kit) and diluted 1:300 in 0.05% Tween 20/PBS, pH 7.2, for 1 h at room temperature. Membranes were washed (×3) in 0.05% Tween 20/PBS, pH 7.2, at room temperature and incubated in ECL™ Plus Detection Reagent (VWR International), and autoradiographic films (Hyperfilm™-ECL, ThermoFisher Scientific) were developed per manufacturer’s instructions. Immunoblot images were captured using ImageJ^®^ Software 1.37v.

### 2.7. Microarray Analysis of Enzymic Antioxidant Gene Expression

Gene expression profiles of patient- and passage-matched OMFs and SFs were assessed using Affymetrix™ GeneChip^®^ Microarray technology, as previously described [9,14]. Briefly, total RNA was isolated from quiescent OMFs and SFs, and RNA extraction was performed according to standard phenol/chloroform extraction protocol. Briefly, first strand cDNA was synthesized from 5 mg total RNA using a T7-(dT)24 primer (Genset Corporation, La Jolla, CA, USA) and reverse-transcribed using a Superscript Double-Stranded cDNA Synthesis Kit (ThermoFisher Scientific). cDNA was purified, and the subsequent in vitro transcription reaction was performed using a Bioarray Kit (Enzo Life Sciences, Exeter, UK) to generate biotinylated cRNA in the presence of T7 RNA polymerase and a biotinylated nucleotide analog/ribonucleotide mix for cRNA amplification and biotin labeling. cRNA was subsequently fragmented and hybridized to Affymetrix™ U133A GeneChips, containing B23 500 sequences derived from the GenBank™ database. Following hybridization and GeneArray^®^ scanning, the resultant image files (.CEL) were analyzed in R, using the Bioconductor RMA package and algorithm to generate expression intensity values for each probe set generated.

### 2.8. Validation of Cellular Superoxide Dismutase 3 (SOD3) Expression Levels

Endogenous SOD3 gene expression in patient-matched OMFs and SFs was assessed using real-time quantitative PCR (RT-qPCR). Patient- and passage-matched OMFs and SFs were established in 6-well plates and maintained at 37 °C in 5% CO_2_/95% air in F-SCM, until confluent. Total RNA was isolated using Qiagen RNeasy Mini Kits (Qiagen Ltd., Manchester, UK) according to manufacturer’s instructions. RNA was quantified using NanoVue™ Plus Spectrophotometer (GE Healthcare, Amersham, UK), and reverse transcription to cDNA was performed using High-Capacity cDNA RT Kits (ThermoFisher Scientific) according to manufacturer’s instructions. Samples, including a negative control (RNA replaced with nuclease-free water), were incubated in a SimpliAmp™ Thermal Cycler (ThermoFisher Scientific) at the following cycle conditions: 25 °C for 10 min; 37 °C for 2 h; 85 °C for 5 min.

RT-qPCR analysis was performed in MicroAmp™ Optical 96-Well Plates using a ViiA™-7 Real-Time PCR System and TaqMan^®^ primers/probe mixes—SOD3 Assay Gene ID: Hs04973910_s1; eukaryotic 18S ribosomal RNA (rRNA) Assay Gene ID: 4310893E, all ThermoFisher Scientific)—according to manufacturer’s protocols. RT-qPCR was performed at a final volume of 20 µL/sample (10 µL TaqMan^®^ Fast Universal PCR Master Mix (2×), 4 µL cDNA, 4 µL nuclease-free water, 1 µL SOD3 primer/probe mix and 1 µL rRNA primer/probe mix) with nuclease-free water replacing cDNA in negative controls. Relative fold changes in SOD3 gene expression (RQ) were calculated using the 2^−ΔΔCt^ method [40], normalized versus an 18S rRNA housekeeping gene.

### 2.9. Validation of Cellular Superoxide Dismutase 3 (SOD3) Protein Levels

Patient- and passage-matched OMFs and SFs were established to confluence, as described above, and harvested with RIPA buffer (ThermoFisher Scientific), containing cOmplete™ mini, EDTA-free Protease Inhibitor Cocktail (Roche), PhosStop™ (Merck) and 2.5 μg/mL sodium orthovanadate (Sigma). Extracts were sonicated, and protein concentrations were quantified as described above. Isolated extracts (10 μg) were subjected to non-reducing SDS-PAGE on pre-formed 4–15% gradient gels, followed by electroblotting onto nitrocellulose membranes (GE Healthcare, Amersham, UK) as described above.

Following electroblotting and blocking, membranes were immuno-probed with primary rabbit anti-superoxide dismutase 3/EC-SOD antibody (Abcam) and diluted 1:1000 in 1% semi-skimmed milk/1% Tween 20 overnight at 4 °C. Normalized protein loading was confirmed using a primary mouse monoclonal anti-GAPDH antibody (1:20,000, Proteintech, Manchester, UK). Membranes were washed (×3) in 1% PBS-Tween and incubated in horseradish peroxidase (HRP)-conjugated swine anti-rabbit polyclonal Ig’s secondary antibody or HRP-conjugated goat anti-mouse polyclonal Ig’s secondary antibody (both 1:5000; Dako, Ely, UK) in 1% semi-skimmed milk/1% Tween 20 for 1 h at room temperature. Membrane washing and protein detection were subsequently performed as described above. Immunoblot images were captured, and densitometry was performed using an Invitrogen™ iBright™ 1500 Imager with iBright Analysis Software Desktop Version v5.1 (ThermoFisher Scientific).

### 2.10. Quantification of Total Cellular Superoxide Dismutase (SOD) Activities

Patient- and passage-matched OMFs and SFs cultures were established, as described above, for assessment of total SOD activities using Superoxide Dismutase Activity Assay Kits (Sigma) according to manufacturer’s instructions. Cells were extracted and lysed using an ice-cold solution of 0.1 M Tris-HCl buffer, containing 0.5% Triton X-100, 5 mM β-mercaptoethanol and 0.1 mg/mL phenylmethylsulfonyl fluoride (all Sigma). Samples, standards (SOD Human Standard, in Kit) and internal assay controls (in Kit) were incubated at 37 °C for 30 min, prior to absorbance values being read spectrophotometrically using a FLUOstar^®^ Omega Plate Reader (BMG Labtech, Aylesbury, UK) at 450 nm. Total SOD activities were subsequently determined versus SOD standard curves, with data expressed as Units/mL.

### 2.11. Statistical Analysis

Microarrays were performed on patient-matched OMFs and SFs isolated from *n* = 4 individuals. All other experiments were performed on *n* = 3 independent occasions. Data were expressed as mean ± standard error of the mean (SEM). O_2_^●−^ generation was analyzed using the Student’s unpaired t-test. SOD3 gene expression, SOD3 Western blot densitometry and total SOD activities were analyzed using the non-parametric Mann–Whitney test. For Microarray data, differentially expressed genes were identified by applying a limma analysis in R, with contrasts defined for the pairwise comparisons between each of our four samples groups. The resulting *p*-values were corrected for multiple hypothesis testing using the false discovery rate (FDR) method [9,14]. Probe sets with an FDR corrected *p* value < 0.05 were annotated and analyzed by overrepresentation analysis, using the Database for Annotation, Visualization and Integrated Discovery (DAVID), Version 6.7 (www.david.abcc.ncifcrf.gov accessed on 14 November 2008). Gene Ontology (GO) biological process categories identified as significant were selected and ranked by Expression Analysis of Systematic Explorer (EASE) score (modified Fisher’s Exact Test). Significance were considered at *p* < 0.05.

## 3. Results

### 3.1. Oxidative Stress Biomarker Detection in Oral Mucosal and Skin Tissues

Representative immunohistochemical images of protein carbonyl group and malondialdehyde detection as respective biomarkers of oxidized protein and lipid damage in patient-matched oral mucosal and skin tissues are shown in Figure 1. Overall, immunostaining for protein carbonyl detection was particularly extensive within the cells and ECM of the oral mucosal lamina propria (Figure 1A,B) and skin dermis (Figure 1C,D) from patient 1, suggesting extensive oxidative protein damage in these regions. In contrast, despite some immuno-positive cells for oxidative protein damage being present, both the oral mucosal epithelium and skin epidermis revealed much weaker immunolocalization for protein carbonyl contents, compared to the lamina propria and dermis. However, no apparent differences in immunostaining were evident between oral mucosal and skin tissues overall. Such patterns of protein carbonyl distribution were consistent for each oral mucosal and skin biopsy analyzed (Appendix A).

In contrast to protein carbonyl profiles, malondialdehyde detection, as a marker of lipid peroxidation, showed greater immunodetection within the oral mucosal epithelium (Figure 1E,F) and skin epidermis (Figure 1G,H) from patient 1, indicating elevated lipid peroxidation in these regions. Although the oral mucosal lamina propria and skin dermis exhibited some staining intensities for malondiadehyde, especially surrounding blood vessels, immunoreactivity was greatly reduced compared to that detectable within the epithelium and epidermis, although the extent of immunostaining in these regions was quite variable across all patient biopsies examined. Specifically, whereas malondialdehyde detection was widespread throughout the oral mucosal epithelium, malondialdehyde immunostaining was only particularly prominent within the stratum granulosum layer of the skin epidermis (arrowed, Figure 1H). However, again, no consistent differences in malondialdehyde immunostaining were evident between oral mucosal and skin tissues, with similar patterns of staining identified for all oral mucosal and skin biopsies analyzed (Appendix A).

### 3.2. Enzymic Antioxidant Detection in Oral Mucosal and Skin Tissues

Representative immunohistochemical images of SOD isoform and catalase detection in patient-matched oral mucosal and skin tissues are shown in Figure 2. Immunohistochemical examination of SOD1 (Figure 2A,B for patient 2) and SOD2 (Figure 2C,D for patient 3) were both more predominantly immunolocalized within the oral mucosal epithelium and skin epidermis, suggesting extensive SOD1 and SOD2 contents in these regions. In contrast, despite containing a low number of SOD1 and SOD2 immuno-positive cells, the oral mucosal lamina propria and skin dermis both demonstrated less staining intensity than the epithelium or epidermis, respectively. The skin epidermis in particular exhibited widespread SOD1 and SOD2 immunostaining throughout (Figure 2B,D), whilst SOD1 and SOD2 were primarily localized to the stratum basale in the oral epithelium (arrowed, Figure 2A,C). However, there were no significant differences in immunostaining between oral mucosal and skin tissues overall, with similar patterns of staining in all oral mucosal and skin biopsies analyzed (Appendix A).

Immunohistochemical analysis demonstrated comparable SOD3 immunodetection within the oral mucosal epithelium and skin epidermis (Figure 2E,F for patient 4). Despite the oral mucosal lamina propria and skin dermis exhibiting comparable diffuse SOD3 immunodetection throughout the ECM, further analysis revealed more intense SOD3 immuno-positive cells within the oral mucosal lamina propria, primarily localized to the cells and blood vessels (arrowed, Figure 2E). Although SOD3 distribution was similar within the dermis, fewer intense immuno-positive regions for SOD3 were apparent than the lamina propria (Figure 2F). Similar patterns of SOD3 distribution and content were evident for all oral mucosal and skin biopsies analyzed (Appendix A).

As with the SOD isoforms, immunohistochemical analysis of catalase distribution was predominantly detectable throughout the oral mucosal epithelium and skin epidermis (Figure 2G,H for patient 3). Although similar patterns of catalase detection were also shown within the oral mucosal lamina propria and skin dermis being particularly localized to the cells, much less catalase immunostaining was apparent within these regions, compared to the intense staining apparent throughout the oral mucosal epithelium and skin epidermis. However, there were no significant differences in immunostaining between oral mucosal and skin tissues overall, with similar patterns of staining in all oral mucosal and skin biopsies analyzed (Appendix A).

### 3.3. Endogenous Reactive Oxygen Species (ROS) Generation by Oral Mucosal and Skin Fibroblasts

ROS generation by OMFs and patient-/passage-matched SFs are shown in Figure 3. Significant increases in superoxide radical (O_2_^●−^) generation by all SF populations (*n* = 4, patients 5–8) were evident at each 24 h, 48 h and 72 h time-point analyzed, compared to OMFs, as quantified by cytochrome C reduction (*p* < 0.001–0.05, Figure 3A). The increased O_2_^●−^ generation by SFs led to further studies comparing overall cellular ROS generation by patient-/passage-matched OMFs and SFs, using DCF and fluorescence microscopy. Images clearly demonstrated increased cellular fluorescence in SFs, versus their patient-/passage-matched OMF counterparts, which only exhibited limited background levels of fluorescence overall (Figure 3B for patient 5; Appendix A for all other patients). On comparison of cellular fluorescence intensities with the corresponding Hoechst nuclear stain images, it was evident that there were more nuclei detected in the OMF cultures, compared to patient-/passage-matched SF cultures. Therefore, this confirmed that there was greater cellular fluorescence and ROS production in SF cultures, despite the presence of fewer cells than OMF cultures.

### 3.4. Oxidative Stress Biomarker Detection in Oral Mucosal and Skin Fibroblasts

Representative immunocytochemical and Western blot images of oxidative DNA and protein damage in OMFs and patient-/passage-matched SFs are shown in Figure 4. Although low intensity cytoplasmic background staining was evident, OMFs exhibited negligible oxidative DNA damage in the form of 8-hydroxy-deoxy-guanosine (8-OHdG) immuno-detection, within the nuclear regions (Figure 4A for patient 5; Appendix A for all other patients). In contrast, SFs clearly demonstrated enhanced detection of DNA fluorescence staining intensities within the nuclear regions, indicative of increased 8-OHdG content and oxidative DNA damage (arrowed, Figure 4A). This was supported by strong co-localization between the detectable 8-OHdG immunostaining (fluorescein isothiocyanate, FITC) and Hoechst nuclear staining (arrowed, Figure 4A), thereby confirming the prominent localization of oxidative DNA damage to the nuclei of SFs.

Western blot profiles for oxidative protein damage, in the form of detectable protein carbonyl content in OMFs and patient-/passage-matched SFs from patients 5 and 7, are shown in Figure 4B (Appendix A for patients 6 and 8). Although all patient-/passage-matched OMFs and SFs exhibited varying levels of detectable protein carbonyl content and oxidative protein damage at many different molecular weights, the profiles generally demonstrated elevated levels of detectable protein carbonyl content at certain molecular weights in SFs, compared to their OMF counterparts. Prominent bands consistently detectable at elevated levels in SF blots for patient 5 were at molecular weights of approximately >200 kDa, 129 kDa, 90–100 kDa, 70–85 kDa, 40–45 kDa, 30–37 kDa and 20–30 kDa (arrowed). Similarly, prominent bands were detectable in SF blots for patient 7 were at molecular weights of approximately >200 kDa, 129 kDa, 80–85 kDa, 45–55 kDa and 30–37 kDa (arrowed).

### 3.5. Enzymic Antioxidant Gene Expression Analysis

As the contrasting oxidative stress biomarker levels identified between OMFs and patient-/passage-matched SFs suggested differences in SOD3 profiles between oral mucosal and skin tissues, we subsequently performed Affymetrix™ Microarray analysis on the OMFs and patient-/passage-matched SFs from patients 5–8 to identify potential differences in enzymic antioxidant expression between these cell populations. However, despite the differences in oxidative stress susceptibilities established, Microarray analysis did not demonstrate any differential expression in enzymic antioxidant genes between OMFs and patient-/passage-matched SFs, such as SOD1, SOD2, SOD3 and catalase (all *p* > 0.05, Table 1). The Affymetrix™ Microarray dataset obtained was deposited on the Gene Expression Omnibus (GEO) database (Accession number, GSE21648) [9,14].

### 3.6. Superoxide Dismutase 3 (SOD3) Expression and Protein Levels in Oral Mucosal and Skin Fibroblasts

Despite the absence of significant differences in enzymic antioxidant expression between OMFs and patient-/passage-matched SFs by Affymetrix™ Microarray analysis, it was intriguing that the lamina propria of oral mucosal tissues exhibited more intensely stained immuno-positive cells for SOD3 than the skin dermis. Thus, we next compared SOD3 gene expression and protein levels between OMFs and patient-/passage-matched SFs to determine whether OMFs were responsible for the elevated SOD3 immunodetection in the lamina propria of oral mucosal tissues.

Relative endogenous SOD3 gene and corresponding protein level expression by OMFs and patient-/passage-matched SFs (*n* = 3, patients 5–7) are shown in Figure 5. Despite RT-qPCR analysis demonstrating that SOD3 expression in OMFs was on average 18-fold higher than their patient-/passage-matched counterparts (Figure 5A), the fold increases in SOD3 expression between OMFs and SFs were deemed to be non-significant overall (*p* > 0.05) due to intrinsic variations in expression between OMFs derived from each patient donor. Such differences led to considerable inter-patient donor variability in OMF SOD3 expression, most notably between Patient 6 (average RQ value = 39.40) versus Patient 7 (average RQ value = 1.52).

Similar findings were evident on comparison of SOD3 Western blot profiles between OMFs and patient-/passage-matched SFs (Figure 5B). All patient-/passage-matched OMFs and SFs exhibited varying levels of detectable SOD3 protein at contrasting molecular weights, with SOD3 particularly detectable in OMFs and SFs at approximately >200 kDa, 130 kDa, 90–100 kDa, 30–37 kDa and 24–30 kDa (arrowed). Such banding patterns are indicative of the presence of octameric, tetrameric and monomeric (uncleaved and heparin-binding domain cleaved) forms of SOD3, respectively [41,42,43]. Such banding patterns are indicative of both intact and proteolytically-cleaved SOD3 subunits, with the mature SOD3 tetramer consisting of cleaved (monomeric) subunits, being detected in particular [43,44,45]. Although SFs demonstrated higher SOD3 immunodetection at high molecular weights than their patient-/passage-matched OMF counterparts for patients 5 and 6 (>200 kDa, 130 kDa and 90–100 kDa), consistent SOD3 immunodetection differences were less apparent at lower molecular weights (30–37 kDa and 24–30 kDa) or between OMFs and SFs derived from patient 7 overall. Consequently, densitometric analyses of SOD3 bands at >200 kDa, 130 kDa, 90–100 kDa, 30–37 kDa, 26 kDa and 24 kDa revealed no significant differences in band intensities between OMFs and patient-/passage-matched SFs (all *p* > 0.05, Figure 5C).

### 3.7. Total Superoxide Dismutase (SOD) Activity Levels in Oral Mucosal and Skin Fibroblasts

The potential contribution of SOD3 to the total cellular SOD activity in OMFs were also compared between OMF and SF populations (*n* = 3, patients 5–7). Overall, no significant differences in total SOD activities were evident between OMFs and their patient-/passage-matched SF counterparts (*p* > 0.05, Figure 6).

## 4. Discussion

The present study sought to examine whether contrasting susceptibilities to oxidative stress exist between patient-matched oral mucosal/skin tissues and OMFs/SFs to ascertain whether superior oral mucosal enzymic antioxidant capabilities and reduced oxidative stress contribute to the preferential healing properties associated with these cells and tissues.

Initial studies demonstrated that similar oxidative stress biomarker and enzymic antioxidant immunostaining patterns were evident in oral mucosal and skin tissues overall. It was further shown that inverse relationships exist between the distributions of elevated oxidative protein damage (in the form of protein carbonyl content) and the lower detection of enzymic antioxidant (SOD1, SOD2, SOD3, catalase) localization within the lamina propria and dermis of the oral mucosa and dermis, respectively. Adaptive enzymic and non-enzymic antioxidant mechanisms are considered critical in maintaining the intracellular redox balance and minimizing ROS-induced oxidative stress, with strong correlations established for enzymic antioxidant upregulation and resistance to cellular oxidative damage [29,30,31,32,33,46]. However, increased oxidative protein damage within the lamina propria and dermis suggest a higher susceptibility to oxidative damage, leading to enhanced protein oxidation or a decreased oxidized protein degradation [47]. Thus, the cumulative effects of extensive ROS exposure could severely impact on protein integrity and the tissue architecture overall, leading to deleterious outcomes for normal cellular repair mechanisms, including OMF and SF wound healing functions.

Although the oral mucosal epithelium and skin epidermis are exposed to the external environment, e.g., salivary peroxidases and ultraviolet light, respectively, the limited protein oxidation in both epithelial tissues is possibly due to the greater enzymic antioxidant capabilities of these epithelia, in agreement with previous reports [48,49,50], and their respective barrier function roles against oxidative insults in these tissues. The differences in enzymic antioxidant profiles between the skin epidermis and dermis have been proposed to be a consequence of the low ratio of cells to ECM within the dermis, which would contribute to the high susceptibility of dermal proteins to oxidative damage, as identified herein [49]. Indeed, the importance of enzymic antioxidants in oral mucosal and skin tissue maintenance is evident from the findings of numerous studies, where despite fibroblasts developing adaptive antioxidant responses to oxidative mediators, reductions or dysfunctions in enzymic antioxidants are well-established as contributors to skin photo-aging and oral/skin carcinogenesis [22,23,24,25,46,49,50,51,52]. As such events occur following ROS exposure during photo-aging and carcinogenesis, it is plausible that such events also occur in skin following exposure to other ROS sources, such as inflammatory cell-derived ROS during impaired chronic wound healing [18,19] and/or dermal fibrosis [20,21]. Indeed, it has been demonstrated that ROS overproduction in chronic wounds results in the inactivation of dermal enzymic antioxidants, despite increased enzymic antioxidant expression in the wound [53,54]. However, although few studies, to date, have addressed such roles for ROS and enzymic antioxidants in oral mucosal wound healing responses, oxidative stress has been implicated in the development of oral submucous fibrosis, an atypical oral mucosal healing condition exhibiting prominent scar formation [34,35].

In contrast to protein carbonyl and enzymic antioxidant profiles, malondialdehyde detection was particularly prevalent within the oral mucosal and skin epithelia, indicative of extensive lipid peroxidation in these regions. Malondialdehyde is formed as a by-product of fatty acid peroxidation, including that associated with the modification and degradation of cell membrane lipids [55]. As such, the predominant detection of malondialdehyde within both oral and skin epithelia suggests a high degree of lipid peroxidation occurs within the keratinocytes of these tissues due to high ROS exposure from their respective external environments. However, in contrast to the scenario with oxidative protein damage, which exhibited limited distribution within oral and skin epithelial tissues as a likely consequence of the elevated enzymic antioxidant capabilities demonstrated above [46,47,48], this was not the case with malondialdehyde. This may be explained by the upregulated enzymic antioxidants in oral and skin epithelial tissues being predominantly intracellular, such as SOD1, SOD2 and catalase, and therefore being potentially less capable of counteracting ROS-induced lipid peroxidation at the cell surface [18,26].

Despite the limited enzymic antioxidant detection in the oral mucosal lamina propria and skin dermis, compared to epithelial tissues, further studies investigated whether inherent differences in oxidative stress susceptibilities existed between patient-/passage-matched OMFs and SFs. OMFs were consistently demonstrated to generate lower levels of ROS compared to patient-/passage-matched SFs. Such findings have previously been attributed to reduced cellular antioxidant status and/or increased ROS production as a consequence of enhanced mitochondrial activity, iron accumulation and NADPH oxidases [56,57,58], although the relative contributions that each mechanism has to the elevated ROS generation in SFs remains to be elucidated. Nonetheless, such increases in ROS generation are further likely to be responsible for the elevated oxidative DNA and protein damage identified in SFs [59,60]. Indeed, oxidative protein/DNA damage and dysfunctional proteosomal/DNA repair mechanisms are established to accompany cellular senescence [61,62]. As previous studies have demonstrated that SFs are more vulnerable to cellular senescence in vitro compared to their patient-matched OMF counterparts due to their possession of “longer” telomeres [8], these findings support the concept that limited ROS generation and their accompanying resistance to oxidative biomolecular damage contribute to OMFs being genotypically and phenotypically “younger” than patient-matched SFs. Such properties, therefore, may be significant in relation to the maintenance of the preferential wound healing and minimal scarring properties of OMFs relative to patient-matched SFs [1,2,4,5,6,7,8,9,10,11,12,13].

These relative susceptibilities to oxidative stress-induced senescence and concomitant increases in cellular ROS generation and oxidative DNA damage would imply that there are distinct differences in the relative antioxidant expression/capabilities of OMFs and SFs. Of the various enzymic antioxidants compared between oral mucosal and skin tissues, increased SOD3 immunolocalization in the oral mucosal lamina propria suggested that it contributed to the maintenance of oxidative balance and, potentially, to the preferential wound healing and minimal fibrosis responses established for OMFs and oral mucosal tissues overall [1,2,4,5,6,7,8,9,10,11,12,13]. However, evaluation of SOD gene expression by Microarray and qRT-PCR, coupled with Western blot protein and activity assay analyses, consistently demonstrated a high degree of variability in detectable SOD3 gene/protein levels between individual patient oral mucosal/skin tissue donors.

SOD3 is particularly localized within the pericellular and ECM environments, in its capacity as the only extracellular scavenger of superoxide radical (O_2_^●−^) species. SOD3, further exhibits greater tissue-specific expression than other SOD isoforms, being prominently expressed in heart, lung and blood vessels [44,45]. Numerous studies have further demonstrated that SOD3 possesses significant protective roles against many oxidative stress-associated diseases, including cancer and various chronic inflammatory conditions, whilst organisms lacking SOD3 are more prone to hyperoxia, resulting in increased disease susceptibilities and shortened lifespans [63,64,65,66,67]. Based on current knowledge of SOD3 involvement in senescence and wound healing, SOD3 is known to reduce telomere shortening rates under both normoxia and hyperoxia, thereby extending fibroblast replicative life-span and protecting against cellular senescence [30,68], whilst SOD3 knockdown is associated with significantly impaired dermal healing as a consequence of decreased fibroblast TGF-β_1_ production and dysfunctional myofibroblast differentiation [69]. Furthermore, analogous studies have demonstrated that elevated SOD3 expression protects against tissue fibrosis characterized by excessive TGF-β_1_ activation, myofibroblast differentiation and collagen deposition [70,71,72,73]. Thus, despite the lack of unequivocal confirmation that SOD3 expression is consistently up-regulated in OMFs derived from all tissue donors, strong similarities exist between the various properties bestowed on fibroblasts that exhibit high SOD3 expression and the preferential wound healing capabilities of OMFs, such as superior proliferation, migration, ECM remodeling, and resistance to cellular senescence and myofibroblast differentiation [1,2,4,5,6,7,8,9,10,11,12,13].

In considering the findings of the present study overall, these suggest that enzymic antioxidants, especially SOD1, SOD2 and catalase, have negligible roles in mediating the limited oxidative stress damage and privileged wound healing responses of OMFs. Therefore, other cellular non-enzymic antioxidant entities may contribute to the preservation of low cellular redox state and wound healing functions in OMFs. Indeed, in addition to its well-established roles in orchestrating fibroblast wound healing responses and alleviating fibrosis [13,74], HGF has also been shown to exhibit antioxidant properties, protecting various cell types against oxidative stress-induced damage [75,76,77]. As HGF expression is highly up-regulated in OMFs [5,9,13,14,15,16,17], it is intriguing to speculate whether HGF contributes similar antioxidant roles to maintain OMF resistance to cellular senescence and scarring and the promotion of its other wound healing activities overall [1,2,4,5,6,7,8,9,10,11,12,13]. Such studies into the potential roles of HGF and other selected non-enzymic antioxidants in regulating OMF wound healing functions will form the basis of our future work in this area.

## 5. Conclusions

Although most enzymic antioxidants were located within oral mucosal and skin epithelia, this study showed that OMFs were more resistant to ROS production and oxidative DNA/protein damage than patient-matched SFs, implying that OMFs possess greater antioxidant capabilities overall. Despite histological evaluation suggesting that the oral mucosal lamina propria possessed higher SOD3 expression, these conclusions were not fully substantiated in OMFs. Such findings suggest that enzymic antioxidants have limited roles in regulating oxidative stress and the privileged wound healing responses of OMFs, implying that other non-enzymic antioxidants may protect OMFs from oxidative stress. Indeed, a greater understanding of antioxidants and their roles in regulating OMF wound healing functions may lead to new strategies to alleviate oxidative stress and improve fibroblast reparative processes in other tissues.

## Figures and Tables

**Figure 1 antioxidants-12-01374-f001:**
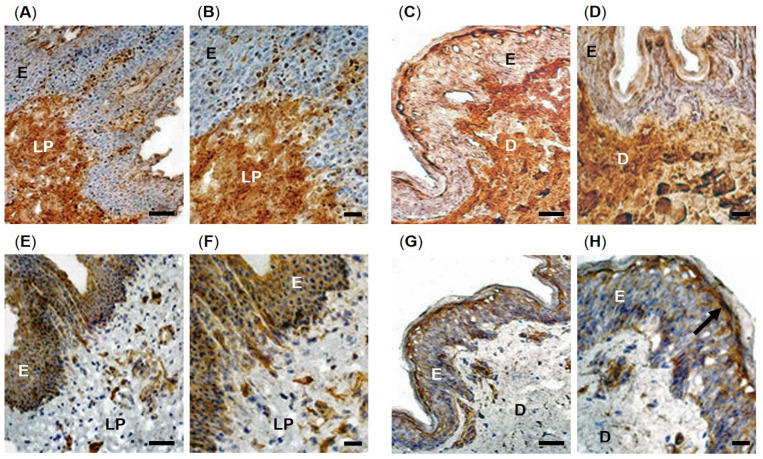
Immunohistochemical localization of oxidative stress biomarkers in patient-matched oral mucosal and skin tissues from patient 1. (**A**) Low and (**B**) high magnification images of protein carbonyl content in oral mucosal tissue. (**C**) Low and (**D**) high magnification images of protein carbonyl content in skin tissue. (**E**) Low and (**F**) high magnification images of malondialdehyde content in oral mucosal tissue. (**G**) Low and (**H**) high magnification images of malondialdehyde content in skin tissue. E = oral mucosal epithelium or skin epidermis; LP = oral mucosal lamina propria; D = skin dermis. Scale bars = 200 μm for (**A**,**C**,**E**,**G**) and 25 μm for (**B**,**D**,**F**,**H**), respectively.

**Figure 2 antioxidants-12-01374-f002:**
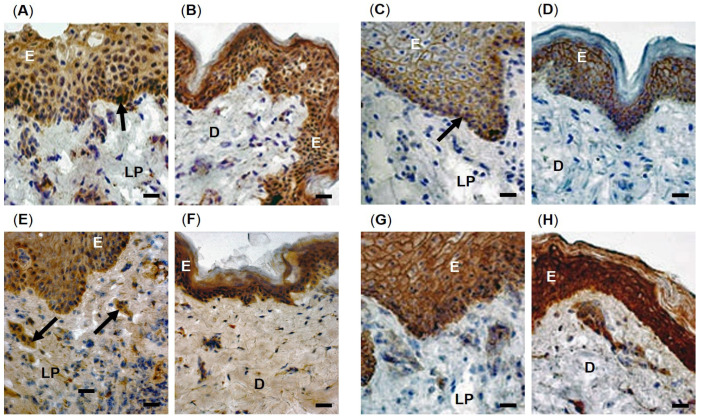
Immunohistochemical localization of enzymic antioxidants in patient-matched oral mucosal and skin tissues. High magnification images of SOD1 detection in (**A**) oral mucosal and (**B**) skin tissues from patient 2, SOD2 detection in (**C**) oral mucosal and (**D**) skin tissues from patient 3, SOD3 detection in (**E**) oral mucosal and (**F**) skin tissues from patient 4, and catalase detection in (**G**) oral mucosal and (**H**) skin tissues from patient 3. E = oral mucosal epithelium or skin epidermis; LP = oral mucosal lamina propria; D = skin dermis. Scale bar = 10 μm.

**Figure 3 antioxidants-12-01374-f003:**
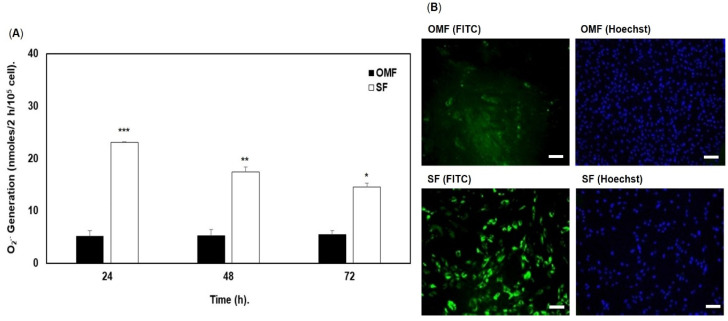
Endogenous ROS generation by patient- and passage-matched OMFs and SFs. (**A**) Mean O_2_^●−^ generation by patient- and passage-matched OMFs and SFs (*n* = 4, patients 5–8), as quantified by cytochrome C reduction over 72 h in culture. Significant increases in O_2_^●−^ generation were evident in SFs at each time-point, compared to OMFs. *n* = 4, values in graph represent mean ± SEM, * *p* < 0.05, ** *p* < 0.01, *** *p* < 0.001. (**B**) Representative FITC (green) and Hoechst nuclear stain (blue) fluorescence microscopy images of DCF detection and ROS generation by patient- and passage-matched OMFs and SFs from patient 5. Scale bar = 200 μm.

**Figure 4 antioxidants-12-01374-f004:**
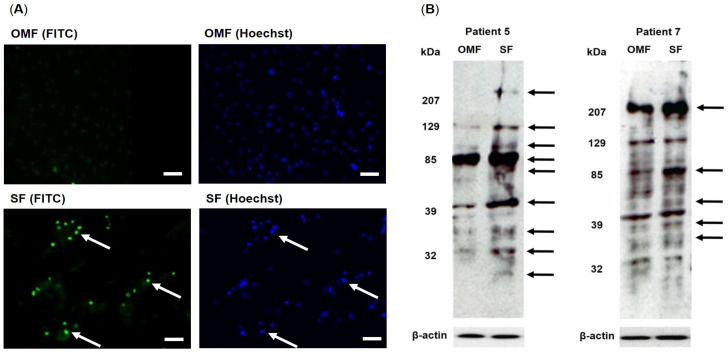
(**A**) Oxidative stress biomarker detection in patient- and passage-matched OMFs and SFs. Representative FITC (green) and Hoechst nuclear stain (blue) fluorescence microscopy images of 8-OHdG detection and oxidative DNA damage in patient- and passage-matched OMFs and SFs from patient 5 (arrowed). Scale bar = 100 μm. (**B**) Representative Western blot profiles of protein carbonyl content and oxidative protein damage in patient- and passage-matched OMFs and SFs from patients 5 and 7. Prominent protein carbonyl bands (arrowed) consistently detectable at elevated levels in SF blots are shown.

**Figure 5 antioxidants-12-01374-f005:**
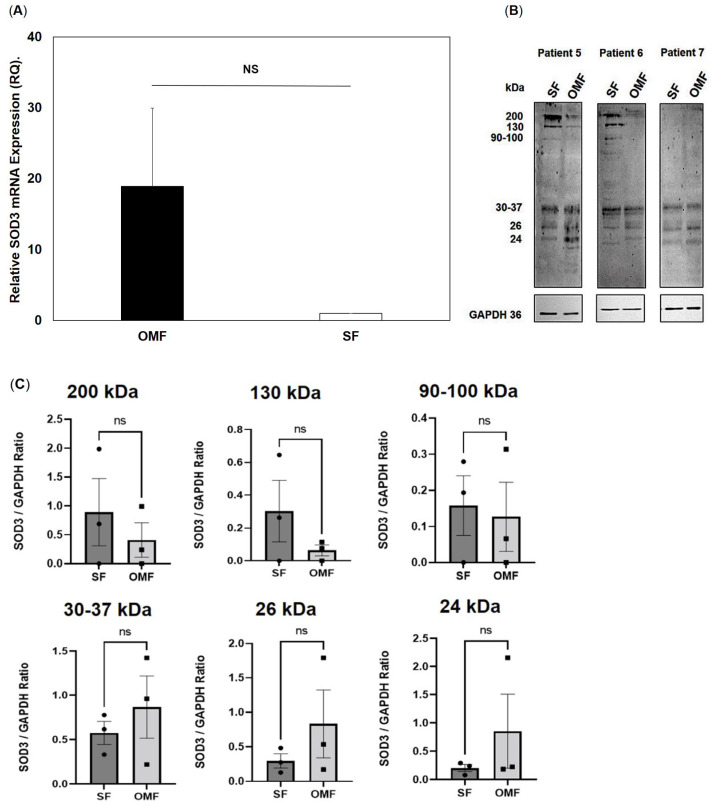
SOD3 gene expression by patient- and passage-matched OMFs and SFs. (**A**) Mean SOD3 expression by patient- and passage-matched OMFs and SFs (*n* = 3, patients 5–7) as quantified by RT-qPCR. Values in graph represent mean ± SEM. No significant differences in SOD3 expression were evident between OMFs and SFs (NS, non-significant; *p* > 0.05). (**B**) Representative Western blot profiles of SOD3 protein contents in patient- and passage-matched OMFs and SFs from patients 5, 6 and 7. Prominent SOD3 protein bands (arrowed) are shown. (**C**) Western blot densitometry for SOD3 band immunodetection at >200 kDa, 130 kDa, 90–100 kDa, 30–37 kDa, 26 kDa and 24 kDa for patient- and passage-matched OMFs and SFs from patients 5, 6 and 7. Values in graph represent mean ± SEM. ns = non-significant, *p* > 0.05. No significant differences in SOD3 protein band densitometry were evident between OMFs and SFs.

**Figure 6 antioxidants-12-01374-f006:**
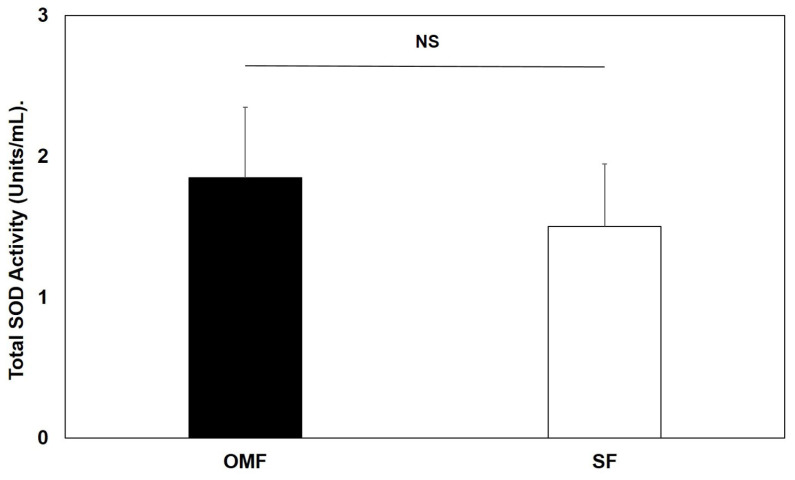
Mean total SOD activities quantified for patient- and passage-matched OMFs and SFs (*n* = 3, patients 5–7). Values in graph represent mean ± SEM. No significant differences in SOD3 activities were evident between OMFs and SFs (NS, non-significant; *p* > 0.05).

**Table 1 antioxidants-12-01374-t001:** Comparison of log_2_-fold differences and significance in SOD1, SOD2, SOD3 and catalase gene expression between patient- and passage-matched OMFs and SFs (*n* = 4, patients 5–8), based on GEO Microarray data. Significance considered at *p* < 0.05.

Enzymic Antioxidant	Affymetrix™ Probe ID	Log_2_-Fold Difference (*FDR p* Value)
SOD1	200642_at	0.078 (0.9996)
SOD2	215078_at	−0.047 (0.9996)
	215223_s_at	−0.336 (0.9996)
	216841_s_at	−0.375 (0.9996)
	221477_s_at	−0.531 (0.9996)
SOD3	205236_x_at	1.706 (0.1848)
Catalase	201432_at	0.342 (0.9996)
	211922_s_at	0.432 (0.9593)
	215573_at	−0.032 (0.9996)

## Data Availability

The data presented in this study are available on request from the corresponding author. The data are not publicly available due to privacy or ethical restrictions.

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
