# Peer review of "Role of Enzymic Antioxidants in Mediating Oxidative Stress and Contrasting Wound Healing Capabilities in Oral Mucosal/Skin Fibroblasts and Tissues"

_antioxidants, 2023, doi:10.3390/antiox12071374_

Round 1

Reviewer 1 Report

The work presented to me for review, entitled " Role of Enzymic Antioxidants in Mediating Oxidative Stress and Contrasting Wound Healing Capabilities in Oral Mucosal/Skin Fibroblasts and Tissues, is an original work. The authors attempt to elucidate the distribution of oxidative stress biomarkers and significant enzymatic antioxidant capacity between patient-matched oral mucosa and skin tissues to determine whether reduced oxidative stress and/or improved enzymatic antioxidant capacity contribute to preferential healing and reduced scarring properties of the oral mucosa. Overall, the topic is interesting and presented in an interesting way. The abstract and the introduction are written correctly with all the key elements undertaken by the author, although I would suggest modify the purpose of the work a little, as it is a bit too complicated. The Materials and Methods is written comprehensively and in some detail. The results are described correctly with attention to detail and all figures presented in the work are added. The authors describe each step of the research in great detail and analyze it thoroughly, which is evidence of great scientific maturity. The discussion section is also correctly written with a correct review of the current literature, although I would remove the first paragraph as it repeats the purpose of the paper from the Introduction. The conclusions are, in my opinion, too long and should rather be reworked and I would rather avoid quotations in the conclusions because they are a summary of the authors' work. The tables and figures presented by the author are adequate to the presented results. Other than that, the article is very interesting and brings new knowledge to the literature.

Minor editing of English language required.

Author Response

Thank you for you comments, which the Authors collectively felt were fair and constructive.  To address your comments:-

1.  As requested by the Reviewer, the latter stages of the Introduction have been amended in an attempt to make this section less complicated (highlighted in yellow, page 2).

2.  Thank you for your suggestion that the first paragraph of the Discussion (Section 4) should be removed (page 13, line 515). However, this very brief statement paragraph was included to reiterate the overall aims of the study to the readers, before commencing the Discussion in full.  Hence, we believe that it would be a good option to retain this brief paragraph at the beginning of the Discussion.

3.  As requested, the Conclusions (Section 5) have now been shorted quite extensively (highlighted in yellow, page 15).

Reviewer 2 Report

This study revealed potential roles of antioxidants in wound healing with focusing on mucosal and skin fibroblasts. Overall, this manuscript is well-written and I have only a few comments.

I am wondering how the OMS and SF samples were “matched”. Similar age with same gender or from same patients?

Fig1G should be upside-down. Epidermis should be upper side. Please change other figures and make the epidermis become upper side for readers to quickly understand the orientation.

Authors should quantify the data of Figure 3 including the cellular fluorescence and cell number.

Authors should add ns in the Figure5A and Figure6 to make it easier to understand more if there are no statistical difference.

Author Response

Thank you for your comments, which the Authors collectively felt were fair and constructive.  To address the Reviewer's comments:-

1.  To confirm, as with some of our previous studies involving comparisons between oral mucosal fibroblasts (OMFs) and skin fibroblasts (SFs) (e.g. Meran et al., J. Biol. Chem. 2007; Enoch et al., J. Dent. Res. 2009, 2010; Dally et al., Int. J. Mol. Sci. 2017), “matched” refers to these fibroblast populations being isolated from the same patients. Information to this effect is included throughout the manuscript, e.g. Abstract, Introduction, Methods, Results and Discussion.

2.  As requested, Fig. 1G and another histological image (Fig. 2A), have now been amended to alter their orientations, so that the epithelial tissues are on the upper regions of the images presented.

3.  Thank you for suggesting that we should quantify the fluorescence data and cell numbers presented in Fig. 3. We assume that the Reviewer is solely referring to Fig. 3B, as the fibroblast superoxide radical (ROS) generation data presented in Fig. 3A (as quantified by cytochrome C assay), demonstrating significantly lower ROS generation by OMFs, versus SFs, are actually normalised for cell number.  Hence, the reason we did not quantify the fluorescence ROS data or calculate the cell numbers was that we already had quantifiable ROS generation data for statistical analysis based on cell number, as presented in Fig. 3A.  Therefore, the additional FITC and Hoechst stain fluorescence images for ROS generation and cell numbers, were only presented as further confirmatory evidence that OMFs generated less ROS than patient-matched SFs (based on the FITC staining intensities), despite OMF cultures containing higher number of cells (based on the Hoechst staining intensities).

4.  As requested, “NS” has now been added to both Fig. 5A and Fig. 6, to denote the non-significant differences between SOD3 expression and activities between patient-matched, OMFs and SFs. Additional text has also been included in the Figure Legends for Fig. 5 and Fig. 6, to explain this (highlighted in yellow, page 12).

Reviewer 3 Report

1. Why cells in the PITC channel and Hoechst channel are not in the same field of view in the ROS generation experiment?

2. DCFH-DA experiment is only a qualitative experiment, a flow-through (quantitative experiment) is recommended to better illustrate the ROS scavenging ability of OMF.

3. In the ROS production experiment, it is suggested that the author controls the same number of cells in SF and OMF, and another group can be added: only cytochrome C is added as a blank control group.

4. Background descriptions for oxidative stress can be strengthened by citing 10.1016/j.cej.2023.141852; 10.1021/acsmacrolett.2c00290 and what are the advantages of the current work compared to published articles?

5.Please clean the background of the protein strip of WB.

6. What is the novelty of this article compared to other articles that have studied rapid healing of the oral mucosa? What problem does this study solve compared to other existing studies?

Author Response

Thank you for your comments, which the Authors collectively felt were fair and constructive.  To address the Reviewer's comments:-

1.  Thank you for your comment. As the Reviewer correctly suggests, we did initially merge the FITC and Hoechst images to produce one image with both fluorescent probe intensities evident.  However, as shown in Fig. 3B, we found it preferable to keep the FITC and Hoechst images separate, in order to more clearly demonstrate that OMFs generated less ROS than patient-matched SFs (based on the FITC staining intensities), even though OMF cultures contained higher number of cells (based on the Hoechst staining intensities).  Unfortunately, this was generally less obvious when we compared our merged FITC/Hoechst images overall, especially at low magnification as we have used.

2.  As we already had quantifiable data on fibroblast superoxide radical (ROS) generation for statistical analysis, as assessed by cytochrome C assay and normalized for cell number (presented in Fig. 3A), we did not pursue any further quantitative analysis of ROS generation between patient-matched OMFs and SFs. To confirm, as detailed above, the additional FITC and Hoechst stain fluorescence images for ROS generation and cell numbers, were only presented as further confirmatory evidence that OMFs generated less ROS than patient-matched SFs.

3.  We are not entirely sure what point the Reviewer is trying to make here. However, as detailed in the Methods (highlighted in yellow, page 3) and as per our previously published paper utilizing this superoxide (ROS) detection method (Wall et al., J. Invest. Dermatol. 2008), all OMFs and SFs are treated with phenol red-free culture medium, containing cytochrome C (80 µM).  Following incubation at 37 °C for 2 h, the culture medium is removed from OMF and SF cultures for the quantification of superoxide radical levels via cytochrome C reduction.  The remaining cells are then counted, in order to normalize superoxide radical generation for cell number, as described above.

4.  Thank you for suggesting the two manuscripts to enhance our descriptions of oxidative stress (i.e. Qi et al., ACS Macro. Lett. 2022; Xiang et al., Chem. Eng. J. 2023). However, having reviewed these manuscripts, both are primarily based upon the development of ROS-scavenging (antioxidant) hydrogels for the treatment of non-healing chronic skin wounds, such as diabetic ulcers.  Although the Xiang paper does utilise a rodent gingival ulcer model as part of the in vivo evaluation of hydrogel antioxidant and healing capabilities, we find limited information or relevance in these papers to the study we have presented, comparing oxidative stress biomarker and enzymic antioxidant profiles between patient-matched oral mucosal/skin tissues and OMFs/SFs.  As such, it is extremely difficult to compare the advantages of our work with these published articles, as the Reviewer suggests.

5.  The Reviewer has requested that we clean the background on our Western blots. However, although the Reviewer did not specify which Western blots they want to improve; we assume this comment relates to those in presented Fig. 5B.  As such, the necessary amendments have been made to the Western blots in Fig. 5B, in an attempt to clean up the background on these images.

6.  As the Reviewer correctly states, and as presented throughout our manuscript (e.g. highlighted in green, page 2), there are numerous published studies describing rapid healing in the oral mucosa and the underlying mechanisms involved. However, even though oxidative stress is a key regulator of wound healing and scarring, and OMFs have been shown to exhibit superior proliferation capabilities, longer telomeres and other desirable wound healing properties than patient-matched SFs (e.g. Meran et al., J. Biol. Chem. 2007; Enoch et al., J. Dent. Res. 2009, 2010; Dally et al., Int. J. Mol. Sci. 2017), prior to our manuscript, no studies had directly compared oxidative stress responses and enzymic antioxidant profiles between oral mucosal and skin cells/tissues.  This information is important, as it enhances of understanding of OMF biology and the mechanisms by which these cells resist oxidative stress and retain their preferential wound healing properties.  We believe that a better understanding of antioxidants and their roles in regulating OMF wound healing functions, may eventually lead to new strategies to alleviate oxidative stress and improve fibroblast reparative processes in other tissues, such as non-healing chronic skin wounds, as mentioned by the Reviewer.